

# The importance of freeze/thaw cycles on lateral tracer transport in ice-wedge polygons

Elchin E. Jafarov[1], Daniil Svyatsky[2], Dylan Harp[1], Brent Newman[1], David Moulton[2], Cathy Wilson[1]

[1]Earth and Environmental Sciences Division, Los Alamos National Laboratory, Los Alamos, NM 87545, USA
[2]Applied Mathematics and Plasma Physics Group, Theoretical Division, Los Alamos National Laboratory, Los Alamos, NM 87545, USA

*Correspondence to*: Elchin E. Jafarov (elchin.jafarov@gmail.com)

**Abstract.** A significant portion of the Arctic coastal plain is classified as polygonal tundra and plays a vital role in soil carbon cycling. Recent research suggests that lateral transport of dissolved carbon could exceed vertical carbon releases to the atmosphere. However, the details of lateral subsurface flow in polygonal tundra have not been well studied. We incorporated a subsurface transport process into an existing state-of-art hydrothermal model. The model captures the physical effects of freeze/thaw cycles on lateral flow in polygonal tundra. The new modeling capability enables non-reactive tracer movement within subsurface. We utilized this new capability to investigate the impact of freeze/thaw cycle on lateral flow in the polygon polygonal tundra. Our study indicates the important role of freeze/thaw cycle and freeze-up effect on lateral tracer transport, suggesting that dissolved species could be transported from the middle of the polygon to the sides within a couple of thaw seasons. Introducing carbon lateral transport in the climate models could substantially reduce the uncertainty associated with the impact of thawing permafrost.

## 1 Introduction

Permafrost stores a vast amount of frozen carbon, which, once thawed, can be released into the atmosphere, amplifying current rates of global warming (McGuire et al., 2018). More than a quarter of the total carbon found in permafrost is stored in tundra ecosystems characterized by organic carbon-rich soil (Strauss et al., 2013). Understanding the impact of permafrost hydrology on the subsurface transport of dissolved carbon is crucial for reducing the uncertainty associated with quantifying the permafrost carbon feedback (Hinzman et al., 2000; Liljedahl et al., 2016; Anderson et al., 2020). A recent study by Plaza et al. (2019) indicates that lateral subsurface flow can be responsible for more than half of the permafrost carbon loss, while many models only consider vertical transport. Because lateral transport of carbon is significant, it is important to be able to understand controls the lateral transport so that the ultimate fate of transported carbon can be ascertained. The rate of carbon transport is not only important for understanding fluxes to lakes and rivers, for example, but transport conditions will also affect how carbon is cycled along a flow path and what form that carbon might ultimately take (e.g., carbon dioxide, methane, or dissolved



organic carbon). Incorporating lateral carbon fluxes into the existing climate models can significantly improve our current understanding and quantification of the permafrost carbon feedback.

In permafrost landscapes, the active layer is the top layer of soil that thaws during the summer. Modeling transport within the active layer remains a challenging problem because of liquid/ice phase change effects on transport. Dissolved constituents including organic carbon, nutrients, redox sensitive species (e.g., iron and methane), mineral weathering products, and

contaminants, are all transported to varying degrees within the active layer during the thaw season and are immobilized when the soil freezes. Early research on transport within the active layer used a one-dimensional approach (Corapcioglu and Panday, 1988; Raisbeck and Mohtadi, 1974). Hinzman et al. (2000) used the SUTRA numerical model to simulate benzene transport at Ft. Wainwright, near Fairbanks, Alaska. That study showed that including permafrost into the simulation strongly influenced the tracer migration pathways. Recently, Frampton and Destouni (2015) applied a coupled subsurface heat and flow model to

study the effect of thawing permafrost on tracer transport under different geological conditions.

Since there was previously no data showing tracer transport in the polygonal tundra, the Next Generation Ecosystem Experiment (NGEE) Arctic team conducted a detailed tracer test on a low- and high-centered polygon at the Barrow Environmental Observatory (BEO) site at North Slope Alaska. The team conducted a two-year tracer experiment and collected the data on transport in polygonal tundra (Wales et al., 2020). Polygonal tundra is characterized by high-, and low-centered

polygon types, with corresponding microtopographic polygon features (troughs, rims, and centers). These surface expressions often follow changes in subsurface structure (i.e., ice wedges and frost table topography), affecting transport pathways. The results of this tracer experiment indicated complex tracer flow in both polygon types, suggesting heterogeneous soil conditions, and a higher lateral rate of tracer flux in the low-centered polygon than in the high-centered polygon. In addition, Wales et al. (2020) found changes in tracer mass occurred over the freeze-up period. The freeze-up period is the time during which the

active layer begins to refreeze until it freezes completely.

The goal of this study is not a direct comparison with the existing data, instead we test the hypothesis using a numerical model postulated as a result of tracer transport observations described in Wales et al. (2020). Wales et al. (2020) suggested that freeze-up could promote additional tracer movement in the active layer of the low- and high-centered polygon. We developed new subsurface transport capability in an existing thermal-hydrology permafrost model to investigate the effect of freeze/thaw

dynamics on the lateral transport of a non-reactive tracer in the polygonal tundra. To test the impact of the freeze/thaw cycle on tracer transport, we setup two model configurations: freeze/thaw disabled and freeze/thaw enabled (Figure 1). The freeze/thaw disabled setup is a simplified version of a permafrost model, where a fixed impermeable layer represents permafrost and there is no ground freezing or thawing. In this setup we run the model only during thaw season (i.e., ground temperatures stay above 0°C). The freeze/thaw enabled setup includes a dynamically moving freeze/thaw boundary and we

run the model over a full-year cycle. We tested these two setups on two main types of polygonal tundra: high- and low-centered polygons. We complemented high- and low-centered polygon simulation with the flat-centered to compare the effect of surface geometry on tracer mobility. This comparison emphasizes the effect of freeze/thaw dynamics on transport in permafrost soils.





We further discussed the effect of subsurface properties on the tracer transport in the polygonal tundra, draw parallels with the experimental study, and describe the possible implication.

## 2 Methods


### 2.1 The Model

We developed a new subsurface non-reactive transport capability in the Advanced Terrestrial Simulator (ATS), a numerical simulator that simulates coupled hydrothermal processes in the permafrost environment (see Appendix). The ATS is a 3D-capable coupled surface/subsurface flow and heat transport model representing the soil physics needed to capture permafrost

dynamics, including the flow of water in variably-saturated, partially-frozen, non-homogeneous soils (Painter et al., 2016). ATS tracks water in its three phases (ice, liquid, and vapor) and simulates the transition between these phases in time. The introduction of tracer transport capability into ATS enables numerical investigation of the processes influencing lateral non-reactive tracer transport in permafrost landscapes, including polygonal tundra, which was not possible previously.

### 2.2 Mathematical Description of the Transport Equation in the ATS

The ATS transport model allows tracer diffusion/dispersion capabilities. However, in this study, we focus on the effect of freeze/thaw on advective transport only. While the diffusion/dispersion processes would lead to additional spreading beyond the advective transport presented here, our results capture the general effect of freeze/thaw on transport in polygonal tundra. The partial differential equations describing advective surface/subsurface transport of the nonreactive tracer have the following form:

$$Subsurface: \frac{\partial(\phi \, \rho_l s_l C_l)}{\partial t} = -\nabla \cdot (\rho_l \boldsymbol{V_l} C_l) + Q^C_{\text{subsurf}} + Q_{\text{surf/subsurf}}, \tag{1}$$

$$Surface: \quad \frac{\partial(\rho_l \delta_l \gamma_l C_l)}{\partial t} = -\nabla \cdot (\rho_l \boldsymbol{U_l} C_l) + Q^C_{\text{surf}} - Q_{\text{surf/subsurf}}. \tag{2}$$

Here, $\phi$ is porosity, $\rho_l$ is liquid density, $s_l$ is liquid saturation, $\delta_l$ is ponded water depth, $\gamma_l$ is effective unfrozen water fraction, $C_l$ is tracer mass, $Q^C$ is a source or sink term of the tracer in the domain indicated by subscript, $Q_{\text{surf/subsurf}}$ represents the mass of the tracer being exchanged from the surface to subsurface. $\boldsymbol{V_l}$ and $\boldsymbol{U_l}$ are subsurface and surface liquid velocities which

are defined by Darcy's law and its surface analog, respectively, as follows:

$$\text{Subsurface liquid velocity:} \quad \boldsymbol{V_l} = -\frac{k_l K}{\mu_l}(\nabla p_l + \rho_l g \boldsymbol{z}), \tag{3}$$



Surface liquid velocity: 
$$\boldsymbol{U}_l = -\frac{\delta_l^{\frac{2}{3}}}{n_{man}(\|\nabla Z_s\|+\varepsilon)^{\frac{1}{2}}}\nabla(Z_s + \delta_l), \qquad (4)$$

where $k_l$ is a relative permeability which depends on the liquid pressure $p_l$. $K$ is the absolute permeability and $\mu_l$ is the liquid
viscosity. $n_{man}$ is the Manning's roughness coefficient and $\nabla Z_s$ is the slope of the surface.

It is a common approach for multiphysics applications to use different time discretization schemes for different physical processes. Since thermal hydrology and transport have different timescale characteristics, it can be advantageous to use an explicit time discretization scheme for the transport processes. In contrast, the fully coupled thermal hydrology model is treated implicitly in time in ATS. An explicit time approximation scheme requires much less computational time than an implicit one,
but to be stable, additional time step restrictions should be imposed. The first-order upwind scheme is applied for explicit discretization of the transport equations (1) and (2). The divergence operator is discretized using integration over the surface of a cell (Gauss theorem). In the first-order upwind scheme, a value of tracer mass is taken from the upwinding cell, with respect to flow velocity, to discretize the surface integral. For this discretization approach, the stable time steps for surface and subsurface domains should satisfy the following conditions:

$$\Delta t_{subsurf} = \min_{cell} \frac{\rho_l s_l \phi}{u_{out}} |v_{cell}|, \qquad (5)$$

$$\Delta t_{surf} = \min_{cell} \frac{\rho_l \delta_l \gamma_l}{u_{out}} |v_{cell}|, \qquad (6)$$

$$\Delta t = \min(\Delta t_{subsurf}, \Delta t_{surf}), \qquad (7)$$

where $u_{out}$ is the total outflux from a particular cell and $|v_{cell}|$ is the volume of the cell.

The selection of time-step criteria depends on the surface flow speed and the amount of liquid saturation in a cell. These
restrictions may lead to expensive and prohibitive computational costs, particularly for long time integration problems. So, it is critical to be able to perform several transport time steps during one hydrology time step, and this procedure is referred to as subcycling. To guarantee mass conservation during subcycling, the interface term should be the same in eqn(1-2). Therefore, the same time step must be used in the surface and subsurface, although the time step restrictions can be different in these domains. The dataset that includes all the ATS input and visualization scripts used in this study is documented in Jafarov and
Svyatsky (2021).

**2.3 Setup**

To investigate the effect of freeze/thaw on tracer transport within polygonal tundra, we created idealized transects of high- and low-centered polygons shown in Figure 1. The 2D transects represent the surface topography and subsurface stratigraphy of



our high- (right plots) and low- (left plots) centered polygons. We set the diameter of the polygons to be the same, 20 m, which

is a representative average value for Arctic polygons (Abolt and Young 2020). We set the elevation of the center of the high-centered polygon 1.25 m above the trough. We set the top of the low-centered polygon rim 50 cm above its center. In both setup cases, the subsurface stratigraphy consists of a peat soil layer overlying a mineral soil layer. In the freeze/thaw disabled case, we used an additional impermeable layer that represents permafrost. The distance between the ground surface and the impermeable layer represents the active layer and equal to 50cm.  In the low-centered polygon, the depth of the peat layer is

approximately 10 cm thick in the center of the polygon and 16 cm thick near the troughs. In the high-centered polygon transect, the depth of the peat layer is uniform and equal to 10 cm. This soil stratigraphy represents the general soil stratigraphy that can be commonly found in polygonal tundra in the Arctic coastal plain around the BEO. The hydrothermal properties for each soil layer were previously calibrated for an ice-wedge polygon at the BEO site and are presented in Table 1 (Jan et al., 2020).

To initialize the ATS model for the freeze/thaw enabled setup, we followed a standard procedure described and implemented

in previous studies (Atchley et al., 2015; Jafarov et al., 2018; Jan et al., 2020). This setup includes the following steps: 1) initialize the water table, 2) introduce the energy equation to establish antecedent permafrost as well as initial ice distribution in the domain, and 3) spin-up the model with the smoothed meteorological data (Jafarov et al., 2020; Jafarov and Svyatsky, 2021). The spin-up is complete when the model achieves a cyclically stable active layer; in other words, when variations in yearly ground temperatures are insignificant. We set the bottom boundary to be a constant temperature of T=-10 ℃ and set

no-flow energy boundary conditions on all other boundaries, except the top boundary. For both types of polygons, a seepage flow boundary condition was prescribed 4 cm below the ground surface on the lateral boundaries of the subsurface domain to allow discharge. These boundary conditions allow water to leave the domain and drainage to the trough network. The final stage of the spin-up procedure was a 10-year simulation of the integrated surface/subsurface hydrothermal model coupled with the surface energy balance and snow distribution model. We used simplified meteorological data to calculate the surface energy

balance that includes a sinusoidally varying air temperature and humidity, constant precipitation, constant radiative forcing, and constant wind. During the thaw season, air temperatures remain above 0 ℃. In our simulations, we used a simple, sinusoidal temperature time-series where the thaw season is 100 days. We applied the tracer 20 days after the first thaw season at the surface of the polygon within a 2-m radius around the polygon center. Here we consider only advective transport. The total mass of tracer injected into the ground was 25.92 mols. The tracer transport occurs within the surface and subsurface

domains, which are coupled via interface tracer fluxes to guarantee the conservation of tracer mass. We assume that the tracer does not mix or become entrapped within ice, so the tracer only resides within the liquid phase.

The freeze/thaw disabled setup was more straightforward and did not include step 2 mentioned in in the above paragraph. First, the initial water table level was established by solving the steady-state variably saturated flow problem. Then we ran the hydrothermal model with an atmospheric temperature of 3 ℃ for ten years. We perform the freeze/thaw disabled simulations

by setting the layer of soil assumed to be permafrost (i.e. annual temperature below 0 ℃) to be impermeable. The depth of the impermeable layer was consistent with the active layer depth for the freeze/thaw enabled condition. The hydrological and thermal parameters used for the peat and the mineral soil layer were the same for the freeze/thaw disabled and enabled setups



(Table 1). We ran this case only during the thaw season, approximating the dynamics of the active layer as a step function. In other words, the entire active layer is thawed instantaneously in the spring and refreezes instantaneously in the fall. In this

setup, at the beginning of the second thaw season we use model outputs from the end of the first thaw season. This setup is not intended to be physical, but is representative of how many hydrothermal permafrost simulations are currently conducted. This setup provides a baseline against which the more physical representation of the freeze/thaw enabled case can be compared.

It can take from several days to weeks or more from the onset of freezing before the active layer fully freezes and, in some cases, it never does fully freeze (Cable et al., 2016). The tracer becomes immobile after the freeze-up period. Therefore, the

time after freeze-up until the next thaw season has no effect on tracer transport. To demonstrate the effect of the freeze-up on the tracer movement, we calculated the fraction of the tracer mass flux that moves within the active layer during freeze-up. We define the beginning of the freeze-up period as the time when the ground surface starts to freeze, encapsulating warmer ground within the active layer in-between the frozen top and the bottom of the active layer.

## 3 Results

Our simulations showed enhanced lateral transport in the freeze/thaw enabled case relative to the freeze/thaw disabled case. In Figure 2, we show the difference between simulated tracer flow for the freeze/thaw disabled and enabled cases for the high-centered polygon at the end of the first and second thaw seasons. After the first year, the tracer in the freeze/thaw disabled case has almost a similar spread to the freeze/thaw enabled case in general (Figure 2a and 2b). After the second year, the tracer amount is lower in the freeze/thaw enabled case as more tracer has left the center due to lateral transport to the troughs (Figure

2c and 2d).

The low-centered polygon simulations showed even more dramatic differences between the freeze/thaw disabled and enabled cases than occurred for the high-centered polygon simulations. For the low-centered polygon freeze/thaw disabled case, more tracer remained in the center of the polygon and vertical transport dominated with little lateral transport (Figure 2e). By the end of the second thaw season, the tracer propagated even deeper (Figure 2g). In contrast to the freeze/thaw disabled case, the

enabled case showed significant lateral transport by the end of the first thaw season (Figure 2f), and by the end of the second thaw season, the differences in lateral versus vertical transport between the two cases were dramatic (Figures 2g and 2h).

To quantify the annual tracer flow rates at different locations from the center of the polygon, we calculated the total tracer mass flow rates (Ftotal) within the active layer for years 1 and 2 (Figure 3). First, we integrated flow rates over the vertical line(face) orthogonally intercepting the polygonal domain, and then integrate those values over a whole year. In the freeze/thaw

enabled case, the low-centered polygon shows tracer mobility up to 6m distance from the center for year 1 and full mobility throughout the polygon for year 2 (Fig. 3a). However, the freeze/thaw disabled case is relatively immobile in comparison, with transport primarily restricted within 2m of the polygon center, which is consistent with Figures 2e and 2g. In the freeze/thaw enabled case, the high-centered polygon shows tracer mobility greater than 6 meters from the center of the polygon for year 1, and full mobility throughout the polygon for year 2. The freeze/thaw disabled case shows similar tracer mobility to the enabled



case for year 1, and much higher mobility for year 2. The tracer is more mobile within 6m distance from the center of the polygon but does not reach the trough even during the second thaw season (Fig 3b), and almost no mobility on the distance from 8 to 10m. Correspondingly, in Figure 2c, at the end of the thaw season 2, there are close to zero concentrations on the sides of the polygon.

In our simulation, the freeze-up period lasts less than two weeks. To illustrate the amount of the tracer moved through the
active layer during the freeze-up period, we calculated the tracer flow rate value during the freeze-up period and divided it by the total flow rate value shown in Figure 3ab. The proportion of tracer flow rate during the freeze-up period relative to the annual flow rate is shown in Figure 3cd (when annual flow rate was greater than 0.001 of tracer mass per year). Between 5% and 40% of the tracer moved through the active layer during freeze up during the first thaw season for the low-centered polygon (Figure 3c). The freeze-up fluxes for the year 1 for the high-centered polygon were less than 10%. During the second year, the
freeze-up proportion of transport was less than 10% in both polygons (Figure 3c and 3d).

# 4 Discussion

## 4.1 Polygon geometry

Wales et al. (2020) field experiment suggests tracer movement during the freeze-up period. We can see that the tracer movement during the freeze-up period is more pronounced for the low-centered polygon (Figure 3c). Differences in high-
centered and low-centered freeze-up effects are likely primarily due to geomorphological differences where the geometry of the polygon dictates the flow of heat (Abolt et al., 2018). For example, high-centered polygons freeze from the center (because it is the highest and most exposed region) and troughs stay relatively warmer. In contrast, the centers of the low-centered polygons stay warmer than their sides due to the cooling effect of their elevated rims (Abolt et al., 2018). Cooler rims should allow water movement towards them. In both cases, during the first year, the freeze-up contributions are greater at the rims
and troughs than in the center. This fact explains the wider spread of the tracer for the freeze/thaw enabled cases. To further test the influence of the polygon geometry, we setup a similar tracer experiment using a flat-centered polygon. Figure 4a shows that simulated annual flow rates for a flat-centered polygon is smaller than similar fluxes for the high- and low-centered polygons. The annual flow rates suggests that none or a very small amount of tracer reaches the trough. The freeze/thaw enabled case, however, moves more tracer than freeze/thaw disabled case for the high-centered polygon. The additional
experiment with the flat-centered polygon shows the importance of the polygon geometry on the tracer movement within the active layer.

## 4.2 Subsurface characteristics

In addition to the polygon geometry, subsurface characteristics can have an impact on tracer mobility. Advective tracer transport is governed by the porosity and permeability of the soil material. So, we further tested tracer mobility by reducing
porosities and permeabilities for the peat and mineral soil layers for the freeze/thaw enabled case for the low- and high-centered



polygonal tundra configurations. Figure 5 corresponds to the reduced porosity case. To differentiate between different cases, we refer to the cases shown in Figures 1 to 3 as the base cases. The annual mass flux simulated for the low-centered polygon (Figure 5a) showed similar dynamics to the base case (Fig. 3c) with the slightly higher mass flux. The annual mass flux simulated for the high-centered polygon (Figure 5b) showed 2 times the mass flux during year 1 and more mass flux towards

the trough of the polygon for year 2 compared to its base case (Fig. 3d). As it was expected, lower porosity reduces storage and therefore increases transport. The corresponding tracer flux during the freeze-up period is reduced by nearly a factor of 2 for both low and high-centered polygons (Figure 5c and 5d) compared to their respective base case scenario (Figure 3c and 3d, respectively).

For the low-centered polygon, reducing soil permeability resulted in a substantial impact on the tracer mobility leading to
reduced annual tracer fluxes (Figure 6a). The comparison between the low permeability run (Figure 6a) and the corresponding base case run (Figure 3c) indicates more than ten times reduction in tracer mobility. The annual mass flux simulated for the high-centered polygon (Figure 6b) was several magnitudes smaller during year 1 and flux was negligible for year 2. For the low-centered polygon, most of the tracer flux occurred during the freeze-up period, suggesting that freeze-up could be a major mechanism moving tracer in low permeable soils. For the high-centered polygon, the tracer fluxes simulated during the freeze-
up period showed more than three times more flux for year 2 than similar fluxes simulated for the base case scenario (Figures 6d and 3d).

### 4.3 Experimental study

Both field and simulations showed consistent freeze-up effect. The main objective of these simulations was not to replicate the field experiment of Wales et al. (2020), because those experiments were conducted on two specific polygons, and the field
polygon sizes, geometeries, stratigraphies, moisture content conditions, etc. do not always correspond to the generalized parameterizations used in the simulations. However, it is worth examining similarities and differences of the field study and the simulations from at least a conceptual viewpoint. The largest difference between the simulations and the field tracer results is that the simulations suggest greater transport in high-centered polygons than low-centered polygons, while the field tracer study indicated the opposite. Wales et al. (2020) suggested that the continuously saturated conditions in the low-centered
polygon (and sometimes ponded surface water conditions) led to greater transport in the low-centered polygon. In contrast, the top of the high-centered polygon where the field tracer was applied was frequently unsaturated. Because the simulations were simplified conceptualizations of high- and low-centered polygons, they may not have fully reproduced the exact conditions during the field test. A future coordinated field and modeling approach would be needed to better constrain transport differences related to polygon types and to refine the simulation approach. The simulations and the field tracer study are
consistent in showing that polygon center to trough lateral transport occurs, that freeze up effects are important, and both the simulations and field experiments show that transport needs to be assessed on a multi-year or multi-thaw season time frame. We define the breakthrough condition when the flux at a specific location is not zero. The simulations and the field tests indicate that more than two years is needed for a full breakthrough of the tracer mass to the trough, and the overall residence



time of the tracer in the polygon is likely quite prolonged, requiring many more thaw seasons. This multi-year time frame has

important implications in terms of actual carbon exports from polygon centers to troughs and likely has substantial effects on biogeochemical conditions, which have been shown to be quite different between polygon centers and troughs (e.g., Newman et al., 2015; Wainwright et al., 2015).

For the low-centered polygon simulations, the differences between the freeze/thaw disabled and enabled cases were more substantial (e.g., Figures 2e-2h and 3a). The freeze/thaw disabled case showed no trough breakthrough over the two-year

simulation period and the freeze/thaw enabled case showed breakthrough in year 2. The field tracer study showed breakthrough in year 1. However, Wales et al., (2020) suggested that this was likely a preferential flow effect which was not considered in the simulations. The freeze-up for the low-centered polygon case only showed a small amount of tracer mass reaching the trough, which is qualitatively consistent with the overall low breakthrough mass observed in the field tracer study.

### 4.4 Cryosuction

One of the key transport considerations during the freeze-up period is cryosuction. During this process, water is drawn towards the freezing front due to capillary forces, a phenomenon that the freeze/thaw enabled simulations include, but the freeze/thaw disabled simulations do not (Painter and Karra, 2014). In the freeze/thaw enabled case simulations, substantial impacts on low-centered polygon transport were observed, consistent with the pre- and post-freeze-up observations of Wales et al. (2020) and supporting their assertion that freeze-up effects may be responsible. For the high-centered polygon, differences between the

freeze/thaw disabled and enabled cases were smaller, suggesting that freeze-up effects were not as important. Apparent freeze-up effects were observed for the high-centered polygon in the field tracer study, but it was difficult to ascertain whether these effects were less pronounced than in the low-centered polygon. Our numerical study illustrates the importance of the freeze/thaw dynamics on the mobility of tracer in polygonal tundra. In some years, the freeze-up period might extend into the late fall due to snowfall and the subsequent subsurface conditions (Cable et al., 2016; Jan et al., 2020). An extended freeze-up

period would further increase tracer movement within a given year, and the multi-year duration for full tracer breakthrough discussed above means that transport would be affected by multiple freeze-up periods. The importance of freeze-up effects on transport may extend beyond polygon areas, affecting a broad range of permafrost landscapes during active layer freezing, and further field and modeling work are warranted to characterize freeze/thaw effects on these other landscapes.

### 4.5 Implications

Monitored ground temperatures in Alaska indicate significant warming over the last 30 years (Wang et al., 2019). Such warming would suggest active layer thickening; however, ground subsidence counteracts the effect of increased temperature on active layer thickness (Streletskiy et al., 2017). Subsidence has hydrological effects due to soil compaction which increases bulk density, and thaw can release previously frozen soil carbon even though apparent active layer thickness does not increase (Jorgenson and Osterkamp, 2005; Plaza et al., 2019). Lateral transport of this carbon would cause redistribution, potentially

affecting local carbon stocks and redox conditions which would affect microbial carbon dioxide and methane production.

Accounting for lateral carbon transport in watershed to earth system scale models would help decrease current uncertainties associated with tundra carbon dynamics. The significance of lateral subsurface transport is highlighted by Plaza et al. (2019), who suggest that more than half of soil carbon loss may be associated with lateral flow. Improved accounting of tundra organic carbon loss can clarify ecosystem source/sink relations and how they may change with additional permafrost degradation. This

study shows that lateral subsurface transport (of carbon and other dissolved constituents) from polygon centers to troughs is important over multi-year timeframes, and that transport during the freeze-up period should be represented in models of these systems, especially in the case of low-centered polygons.

## 4 Conclusions

The numerical experiments we conducted in this study illustrate the importance of polygon type and the freeze/thaw cycle on

transport. We showed that not considering the freeze/thaw cycle while modeling hydrological and carbon fluxes in polygonal tundra could be misleading. In particular, in the low-centered polygons, not including the freeze/thaw cycle led to tracer immobility. On the other hand, the high-centered polygon simulations for the freeze/thaw disabled case were similar to the tracer mobility for the freeze/thaw enabled case during the first thaw season. However, simulated total tracer fluxes for the second thaw season showed different tracer mobility (Figure 3b). For the freeze/thaw disabled case, despite high fluxes at 4

meters from the polygon center, the tracer failed to reach the trough. Thus, tracer movement during the freeze-up period could have a significant impact on tracer mobility. The freeze-up effect is more pronounced for the low-centered polygons and for a low permeability soil, freeze-up could be a major mechanism transporting tracer within the polygon.

## Acknowledgments

This work is part of the Next-Generation Ecosystem Experiments (NGEE Arctic) project which is supported by the Office of

Biological and Environmental Research in the DOE Office of Science.

## Appendix

The ATS is a collection of representations of physical processes designed to work within a flexibly configured modeling framework (Moulton et al., 2012; Coon et al., 2016). This modeling framework is a powerful multiphysics simulator designed on the basis of the Arcos management system (Coon et al., 2016). In this study, we coupled the surface and subsurface thermo-

hydrology and surface energy balance components of ATS (Atchley et al., 2015; Painter et al., 2016; Jafarov et al., 2018) with a nonreactive transport model. This configuration of ATS is an integrated thermo-hydrologic model including a surface energy balance, enabling the simulation of snowpack evolution, surface runoff, and subsurface flow in the presence of phase change.



The entire system is mass and energy conservative and is forced by meteorological data and appropriate initial and boundary conditions.

The surface energy balance model uses incoming and outgoing radiation, latent and sensible heat exchange, and diffusion of energy to the soil to determine a snowpack temperature and simulate the evolution of the snowpack (Atchley et al., 2015). The snowpack evolution model includes snow aging, density changes, and depth hoar and, based on subsurface conditions, determines energy fluxes between the surface and subsurface. Overland flow is modeled using the diffusion wave approximation, which is derived based on Manning's surficial roughness coefficient, with modifications to account for frozen

surface water. Additionally, transport of energy on the surface is tracked through an advection-diffusion transport equation that supports phase changes.

Subsurface thermal hydrology is represented in ATS by a modified Richards' equation coupled to an energy equation. Unfrozen water at temperatures below 0oC, an important physical phenomenon for accurate modeling of permafrost (e.g., Romanovsky and Osterkamp, 2000), is represented through a constitutive relation (Painter and Karra, 2014) based on the

Clapeyron equation, which partitions water into the three phases. Density changes associated with phase change and cryosuction (i.e., the "freezing equals drying" approximation) are included. Karra et al. (2014) demonstrated that this formulation is able to reproduce several variably saturated, freezing soil laboratory experiments without resorting to empirical soil freezing curves or impedance factors in the relative permeability model.

Other key constitutive equations for this model include a water retention curve, here the van Genuchten–Mualem formulation

(van Genuchten, 1980; Mualem, 1976), and a model for thermal conductivity of the air-water-ice-soil mixture. The method for calculating thermal conductivities of the air-water-ice-soil mixture is based on interpolating between saturated frozen, saturated unfrozen, and fully dry states (Painter et al., 2016), where the thermal conductivities of each end-member state is determined by the thermal conductivity of the components (soil grains, air, water, or liquid) weighted by the relative abundance of each component in the cell (Johansen, 1977; Peters-Lidard et al., 1998; Atchley et al., 2015). Standard empirical fits are

used for the internal energy of each component of the air-water-ice-soil mixture.

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

**Table 1.** Subsurface thermal properties used for low- and high-centered polygons (from Jan et al., 2020).

| Parameter | Peat | Mineral | Impermeable |
|---|---|---|---|
| Porosity | 0.87 | 0.56 | 0 |
| Intrinsic permeability [m$^2$] | $5^{-10}$ | $2^{-11}$ | $5^{-20}$ |
| Residual water content [-] | 0 | 0.2 | 0 |
| van Genuchten $\alpha$ [1/m] | $2.93 \cdot 10^{-4}$ | $3.3 \cdot 10^{-4}$ | $5.45 \cdot 10^{-4}$ |
| van Genuchten $m$ [-] | 0.212 | 0.248 | 0.191 |
| Thermal conductivity ($S_{uf}$) W/(mK) | 1.0 | 1.1 | 1.0 |
| Thermal conductivity (dry )W/(mK) | 0.29 | .3 | 0.29 |





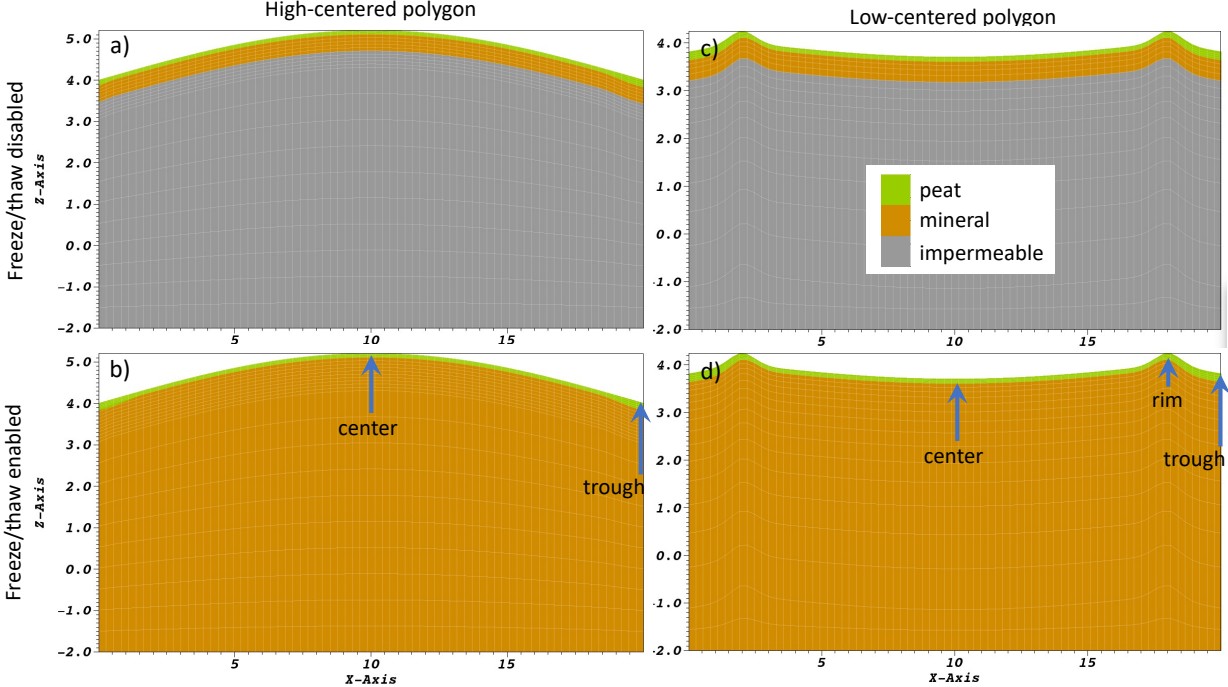

**Figure 1.** ATS meshes representing 2D transects of a high-centered polygon (a and b) and a low-centered polygon
(c and d). The top row illustrates the stratigraphies representing the freeze/thaw disabled case and the bottom row
the freeze/thaw enabled case. The green represents peat, brown the mineral soil layer, and grey the impermeable
layer of the freeze/thaw disabled case.





**Figure 2.** Snapshots of the tracer simulation for the freeze/thaw disabled (a, c, e, and g) and freeze/thaw enabled (b, d, f, and h) cases for the high-centered polygon. The top 4 plots are for the high-centered polygon and the bottom 4 are for the low-centered polygon. The first row (a, b) and the third row (e, f) indicate the end of the first thaw season and the second row (c and d) and the fourth row (g and h) indicate the end of the second thaw season. The colorbar represents the fraction of tracer mass with respect to the total mass injected. Here we do not show $m/m_{total}$ less than $10^{-5}$.







**Figure 3**. The total flow rate ($F_{total}$) represents the tracer flow rates integrated over the vertical cross-section of the polygon and over the entire year at different distances from the center of the a) low-centered and b) high-centered polygon. The proportion of tracer flow rate during the freeze-up period relative to the total flow rate at different distance from the center of the c) low-centered and d) high-centered polygon. $F_{freezeup}$ represents the tracer flow rates integrated over the vertical cross-section of the polygon and over the freezeup period at different distances from the center. The solid red curve corresponds to the year 1 and the solid blue corresponds to the year 2 total flow rates for the freeze/thaw enabled case. The dashed red curve corresponds to the year 1 and the dashed blue corresponds to the year 2 total flow rates for the freeze/thaw disabled case.





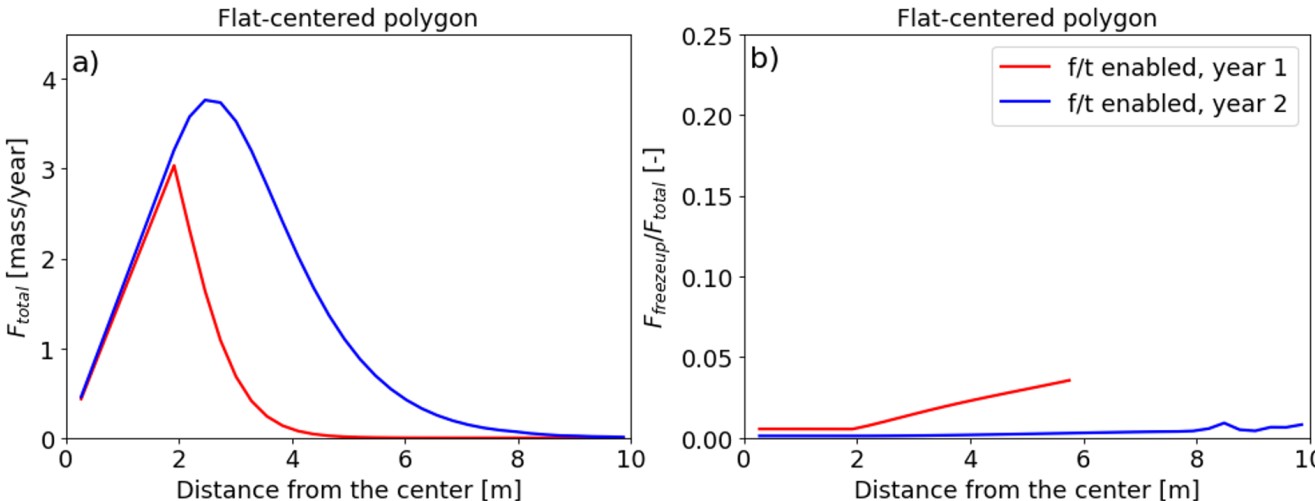

**Figure 4.** (a) The total flow rate ($F_{total}$) represents the tracer flow rates integrated over the vertical cross-section of the flat-centered polygon and over the entire year at different distances from the center. (b)The proportion of tracer flow rate during the freeze-up period relative to the total flow rate at different distance from the center of the flat-centered polygon. $F_{freezeup}$ represents the tracer flow rates integrated over the vertical cross-section of the flat-centered polygon and over the freezeup period at different distances from the center.

455





**Figure 5.** The low porosity setup. The total flow rate ($F_{total}$) represents the tracer flow rates integrated over the vertical cross-section of the polygon and over the entire year at different distances from the center of the a) low-centered and b) high-centered polygon. The proportion of tracer flow rate during the freeze-up period relative to the total flow rate at different distance from the center of the c) low-centered and d) high-centered polygon. $F_{freezeup}$ represents the tracer flow rates integrated over the vertical cross-section of the polygon and over the freezeup period at different distances from the center.





465

**Figure 6.** The low permeability setup. The total flow rate ($F_{total}$) represents the tracer flow rates integrated over the vertical cross-section of the polygon and over the entire year at different distances from the center of the a) low-centered and b) high-centered polygon. The proportion of tracer flow rate during the freeze-up period relative to the total flow rate at different distance from the center of the c) low-centered and d) high-centered polygon. $F_{freezeup}$

470 represents the tracer flow rates integrated over the vertical cross-section of the polygon and over the freezeup period at different distances from the center.