# Peer review of "The importance of freeze/thaw cycles on lateral tracer transport in ice-wedge polygons"

_The Cryosphere, 2021_

## Author Comment (AC1)

We greatly thank our referees for their careful review and valuable comments, which significantly strengthen our manuscript. We made substantial changes to the manuscript. We moved a part of the discussion into results, extended the discussion section based on the reviewers' comments, and rewrote the conclusion. We have modified the manuscript in response to these comments and elaborate below on our responses (in dark blue) to their comments.

**RC1**

Jafarov et al. conduct a modeling study using ATS to assess the impact of freeze-thaw and freeze-up on lateral flow in high- and low-centered polygons. This study integrates a new modeling capability into ATS that enables tracking of non-reactive tracer movement due to advective transport. This study incorporates two transects (i.e., domains) representing high- and low-centered polygons and two freeze thaw scenarios – one scenario with seasonally dynamic freeze-thaw and one scenario with the active season only (i.e., excluding freeze-thaw and freeze-up) to assess lateral tracer movement over two years. This study confirmed field studies by Wales et al. (2020) that suggests that freeze-up has a large impact on tracer transport. Simulations show significant lateral transport within two thaw seasons and this transport was greater in scenarios with freeze-thaw. Results also showed that the impact of freeze-thaw was greater in low-centered polygons, as evidence by the increased difference in tracer transport between freeze-thaw enabled and disabled scenarios for low-centered polygons compared to high-centered polygons.

Overall, this is an interesting and important study and will be of interest to the readership of The Cryosphere. This study not only contributes important advances in cryohydrogeological modeling, but also provides important insights into lateral flow in ice-wedge polygons and potential carbon transport. The modeling setup and analyses are sound, and the results are clearly conveyed.

I have a few general and line specific comments articulated below:

**General comments:**

• The discussion section needs to be expanded and developed to include more explanation and context to the points addressed. In select sections such as 4.3, the discussion reads more like a results section than a discussion of the findings, leaving the reader with questions such as: Why does freeze-up impact tracer transport? Why is it greater in some scenarios than others? What are the drivers and processes at play? Why does most of the tracer flux for low-centered polygons occur during the freeze-up period in low permeability scenarios? The results are interesting and important, but the manuscript lacks depth beyond results reporting. Addressing these questions in addition to other similar questions will greatly improve the manuscript and the impact of the paper.

**We added the paragraph that answers the above questions into the Discussion (L279-320).**

The freeze-up impacts tracer transport as a result of thinning of the unfrozen space between the top and bottom freezing fronts of the active layer, pushing tracer further to the sides. The polygon geometry and subsurface characteristics could amplify or reduce the freeze-up impact. For example, low-centered polygon geometry allows more tracer transport than flat-centered geometry under the same meteorological inputs. On the other hand, tracer mobility depends on subsurface permeability and porosity characteristics, slowing down or accelerating tracer transport. We showed that reducing porosity and permeability reduces tracer mobility amplifying the impact of freeze-up. Suggesting that closing freezing fronts push more tracer during freeze-up. This effect is a result of the Claus-Clapeyron equation when liquid is pushed from thawed to frozen parts of the ground described by Davis (2001).

• Similarly, the drivers of transport were not discussed in the manuscript. There is no text discussing hydraulic gradients and little with respect to thermal gradients. This is important when discussing lateral flow with time. How do the hydraulic gradients change with time? Between scenarios? How about the thermal gradients? I strongly addressing these questions, or thoroughly discussing drivers of flow in the text. Adding in even a few well-crafted and well-placed sentences discussing the processes influencing the results will significantly improve the manuscript and its impact.

We added the following paragraph into the Discussion (L321-325).

Our results illustrate the importance of hydraulic and thermal gradients. The impact of hydraulic gradients plays an essential role for the high-centered polygons. The disabled freeze/thaw case (Fig 2a and 2c) confirms the strong influence of hydraulic gradients between center and troughs of the high-centered polygon, providing almost similar tracer mobility simulated for the freeze/thaw enabled case. Inversely, thermal gradients move tracer in the case of the low-centered polygon, indicating that without thermal gradients, the tracer would stay immobile (Fig 2e and 2g).

• While a tracer is one way to track lateral flow of a non-reactive solute, DOC is reactive. Given the focus on and links to carbon transport made in the manuscript, I suggest adding text that addresses how DOC transport differs from tracer transport. Perhaps it is obvious, however, it is important to mention that DOC is (1) reactive, (2) can be entrapped in ice, and (3) impacted by local conditions. Please also add text as to how this may impact the results of this study.

Agree, we have added the text below starting at line 87 of the manuscript. We also included additional information on transport effects in our response to the review comment for line 30.

It is important to note that this study focuses on non-reactive "tracer" transport because this is an important basis for characterizing polygon hydrology. However, it does not represent reactive transport conditions that are critical for modelling solutes such as DOC or nutrients. In other words, the non-reactive transport models described here are a step toward more sophisticated models that include processes such as changing source contributions along a flow path, dissolution/precipitation, sorption phenomena, and microbial/redox processes.

• Please check the manuscript for typos. There are several instanced throughout the manuscript, some of which have been identified below.

**We corrected the typos.**

**Specific comments:**

Line 26-27: Rephrase. Perhaps add 'on' after 'controls'. Done

Line 28: I suggest adding examples of transport conditions.

Agree, we have added the following to line 30:

A review by O'Donnell et al. (2021) highlights several studies that demonstrate a variety of Arctic flow and transport conditions that affect dissolved organic carbon (DOC) and nutrients such as nitrogen and phosphorous. For example, DOC fluxes typically increase during spring snowmelt where substantial lateral flushing occurs. After snowmelt, DOC fluxes tend to decrease especially when active layers deepen into mineral soil layers. DOC compositions also shift as lateral flow through mineral soils increases or mixing with deeper groundwater sources becomes more important. Microbial utilization of DOC and nutrients can change concentrations along hydrological flow paths, and if redox conditions vary with depth or laterally, profound changes in concentrations and/or the geochemical forms of carbon (and nutrients) can result.

Line 45-55: This section is repetitive and seems to jump between sentences. I suggest reordering the text to incorporate the hypothesis in the first paragraph then what was done in the second to avoid repeating findings of Wales et al. (2020). Great suggestion. We implemented it as suggested.

Line 51: add 'but' after the comma. Done

Line 61: Add an 's' to simulation. Done

Line 64: Add an 's' to implication. Done

We added more clarification on the terms and units of the variables that participated in the transport equations (L110-126), which also address two comments below.

Line 83: Please clarify the Qc term and how it is determined. tracer molar fraction

Equation 4: Define  $\varepsilon$ .  $\varepsilon$  is a regularization parameter that helps to deal with zero gradient slopes.

116: Please add a reference for the domain specifications. We include a clarification "... based on our observations at the BEO site".

Line 199. Please add 'is' after 'and'. Done

Line 130. For clarity, this sentence could be rephased to 'no-flow energy boundary conditions on the sides of the domain.' Switching between boundaries and sides could confuse readers, for consistency we left 'boundaries'.

Line 131: Please more thoroughly explain the 4 cm. Why at 4 cm? Is this a point or a line segment, and if so, from where to 4 cm? Would the seepage location impact the results? The seepage condition allows snowmelt water to leave the domain and prevents it from pooling at the surface. Pooling water would lead to absorption of the radiative heat at the surface affecting active layer thickness. A specific depth (4 cm below the surface) was chosen based on the assumption that the active layer at the beginning of the snowmelt is small and 4cm stands as a conservative estimate.

Line 132: I believe 'drainage' should be 'drain'. Done

Line 137: I suggest adding 'started' after 'season'. Done

Line 142: Remove the second 'in'. Done

Line 145-150: Were diurnal temperature changes included? No diurnal temperature changes were included.

Line 257: This sentence needs more detail. Impacts of what? "may be responsible…" for what? More text in this sentence will add clarity and make the point (and this section) clearer. We added a missing text "...for further tracer transport".

Figure 1: Please include units in the axis titles. Also, I suggest reassigning letters to the plots to match the rest of the figures (i.e., a,b,c,d rather than a,c,b,d). Additionally, it may help to find a way to indicate a dynamics permafrost boundary in b and d. The depiction as is was initially confusing. Done

**RC2**

The paper discusses the effect of freeze/thaw cycles on lateral transport of conservative solutes in polygonal tundra, and links this transport to soil carbon cycling. A new transport capability is incorporated into ATS, an existing hydrothermal model. By comparison to field data, the modelled cases indicate that lateral transport may be an important feature of dissolved carbon.

The paper is well structured, easy to follow and the modelling mostly appears scientifically sound and well performed.

Below, I first provide some more general comments, followed by minor or editorial comments.

Major comments:

1. While the introduction nicely explains why lateral transport of carbon is important to understand and quantify, it remains somewhat unclear if the lateral transport in a single polygon is critical. That is, the paper could somewhat elaborate on what happens with dissolved carbon once it reaches the edge (trough) of a single polygon. If it is released immediately similarly to the carbon which is transported vertically, maybe the effect is minor. Also, it would be helpful if the authors could provide some more detail on in what form carbon is transported, and what its ultimate form is when released. Maybe by outlining the relevant reactions this would become clearer.

Please refer to the third comment of RC 1. Introducing chemical reactions is beyond the scope and out of focus of the current study.

2. In order to mimic the tracer experiment, the tracer source is placed on the surface of the polygon. I wonder if not a more realistic source, in order to ultimately understand DOC transport, would have been to distribute the source vertically within the active layer.

Here, we have been following the actual experiment design. Distributing tracer equally within the active layer would be hard to do experimentally due to heterogenous soil surface and subsurface characteristics. Numerically we could design such an experiment, but it is out of the scope of the existing study.

3. The paper clearly demonstrates how the freeze/thaw cycle promotes lateral transport relative to the freeze/thaw disabled case. But is not this enhanced transport mainly a fact of assigning a zero permeability in the active layer for the freeze/thaw disabled case such that solute becomes immobilized for large parts of the year? That is, how does vertical transport compare between the two cases; is that also enhanced for the freeze/thaw enabled case?

Permeability is not zero in the active layer for all cases simulated in this study. Zero permeability would represent completely frozen material. The solute is immobilized in Fig 2e and 2g due to the absence of the hydraulic forces pulling tracer to the sides, as shown in Fig 2a and 2c. We also expanded on this topic on a similar comment raised by a previous reviewer (first and second comments RC1). The vertical transport is controlled by the head pressure difference between center and troughs. The corresponding text is added to the Discussion section (4.2) of the manuscript.

- 4. Concerning the governing equations (1)-(2) and equation (3), I note the following:
  - 1. Cl should be mass fraction (rather than mass as stated)? It is tracer molar fraction.
  - 2. The units of the LHS of eq (1) and (2) are not the same (seem to differ by a length unit). We added more clarification on the terms and units of the variables that participated in the transport equations (L110-126), which also address two comments below.

- 3. .In eq (3), should not the RHS be divided by porosity in order to obtain the liquid velocity? No, it is the definition of Darcy flux.
- 5. Did Wales et al. (2020) observe the same pattern (differences) between the two consecutive years as in the numerical simulations? If not, please elaborate on the possible deviation.

We added an additional text into the Discussion section (4.3).

In the experiment conducted by Wales et al., (2020) tracer concertation dropped for year two and did not resume where it ended in the previous year. Most of the tracer traveled vertically over the first year reached the bottom of the active layer. The mobility did not resume until the end of the second thaw season when the active layer was at its maximum depth. Similar behavior was exhibited by the simulated tracer. The heterogeneous active layer depth within the actual polygon in the field experiment could lead to different pathways found by the tracer and further tracer distribution and overall dilution. Our numerical experiment is more representative of the ideal lab experiment than the actual field study and, therefore, not directly comparable with the complex conditions in the field. The main objective of these simulations was not to replicate the field experiment of Wales et al. (2020), because those experiments were conducted on two specific polygons, and the field polygon sizes, geometries, stratigraphies, moisture content conditions, etc. do not always correspond to the generalized parameterizations used in the simulations. However, it is worth examining similarities and differences of the field study and the simulations from at least a conceptual viewpoint.

6. As a complement to Figures 3-6, I think it would be helpful to plot cumulative mass discharge as a function of time at the polygonal boundary (i.e., at 10 m distance from center). This would aid in understanding the temporal evolution.

For the illustrative purpose, we plotted the drainage for the freeze/thaw-enabled polygons shown in Figure 2. We used the constant average summer and winter precip. As seen from the plots, the discharge is increasing over summertime and zero during the wintertime, as expected. We do not see a substantial value by plotting discharge for all cases. Therefore, we decided to include them in the manuscript.

Minor comments:

- 1. Line 27: insert 'what' before 'controls'. We changed it to "to understand controls on the lateral transport"
- 2. Line 119: Should be 'equals'. Done
- 3. Lines 137-138: Clarify when the tracer is applied; i.e., is it 20 days after the end of the first thaw season? We added the clarifying statement: "We applied the tracer 20 days after the first thaw season started ..."
- 4. Lines 145-146: Is the impermeable layer constant in space? The impermeability layer follows the geometry of the specific polygon type shown in Fig 1a and 1b. The active layer depth is constant everywhere within the transect.
- 5. Line 153: Specify if the suggested time period of several days to weeks is site-specific or a general statement. A general statement. We included the following clarification: "In general, ..."
- 6. Line 211: How much is porosity reduced in this variant? We added a clarification in the text for the low porosity case porosity was reduced by a factor of 2. L251
- 7. Line 219: How much is permeability reduced in this variant? We added a clarification in the text for the low permeability case permeability was reduced by a factor of 10. L254
- 8. Lines 235-238: Did the modelling include unsaturated flow? No, it did not. In general, soils in this part of the Arctic stay saturated over the entire summer.
- 9. Line 313: Superscript missing in 0°C. Done